# Ultra-marginal Feature Importance

## Abstract

Scientists frequently prioritize learning from data rather than training the best possible model; however, research in machine learning often prioritizes the latter. Marginal contribution feature importance (MCI) was developed to break this trend by providing a useful framework for quantifying the relationships in data in an interpretable fashion. In this work, we aim to improve upon the theoretical properties, performance, and runtime of MCI by introducing ultra-marginal feature importance (UMFI), which uses preprocessing methods from the AI fairness literature to remove dependencies in the feature set prior to measuring predictive power. We show on real and simulated data that UMFI performs better than MCI, especially in the presence of correlated interactions and unrelated features, while partially learning the structure of the causal graph and reducing the exponential runtime of MCI to super-linear.

## 1   Introduction

Scientists often seek to determine the true relationships between a set of characteristics and some outcome of interest. These relationships are ideally determined by performing carefully controlled experiments so that causality can be established. However, experiments can be difficult and costly to pursue, unethical to perform, or impossible to control [51, 44], leaving only observational data available. The relationships that are hidden within vast quantities of observational data are often difficult to determine, so statistical tools, such as feature importance, have been explored.

Recently, feature importance methods such as Shapely-values [40, 13, 33], SAGE [14], accumulated local effects (ALE) [3], permutation importance (PI) [8], and conditional permutation importance (CPI) [16] have been used in high-impact journal papers by scientists who want to explain the mechanisms within data [2, 5, 42, 29, 38, 19, 26]. However, these methods may not adequately explain data in certain circumstances [12, 11]. ALE can only easily show first order effects [36], and although CPI improves upon some limitations of PI, CPI has the property that two perfectly correlated features with significant predictive power would both be deemed unimportant [14]. Further, only one model is trained in ALE, CPI, and PI. Thus, correlated features, which can alter the model assembly process, could be given artificially low importance if the goal is to explain the data [24]. Instead of exploring a single model, the developers of SAGE, SPVIM, and marginal contribution feature importance (MCI) evaluate the difference in accuracy between a model trained with the feature of interest and a model trained without it, across all feature subsets [11, 14, 49], though these methods are prevented from being accepted by a wider scientific audience because of their high computational cost. In particular, we note that MCI is the current state-of-the-art method for explaining data as it was shown in extensive experiments to have better quality and robustness when compared to Shapely-values, SAGE, ablation, and bivariate methods [11].

Though MCI can be seen as the current state-of-the-art method for explaining the data, it has three key shortcomings. First, the exact computation of MCI requires an exponential number of model trainings, which makes MCI ineffective at interpreting large datasets (e.g., gene expression studies).

Second, although it can handle complex feature interactions and data with correlated features, MCI underestimates the importance of correlated features that form interaction effects because MCI usually ignores features that share information with the feature of interest $x_i$. Even if $x_i$ and $x_j$ form an interaction effect, the additional predictive power offered by $x_i$ on top of a subset $S$ would be diminished by the presence of $x_j \in S$, provided that the correlation between $x_j$ and $x_i$ is strong enough. Third, MCI can give non-zero importance to features that are completely unrelated to the response variable, as experimentally shown in Catav et al. [11, Figure S3] and theoretically shown in Harel et al. [23]. We hypothesize that constructing independent and information-preserving representations of the data could resolve these three issues. With this in mind, we introduce ultra-marginal feature importance (UMFI), a new variable importance method that can better explain the data while drastically reducing runtime.

The rest of this paper is organized as follows. Axioms for explaining the data are proposed in Section 2. The framework for UMFI is then formally presented in Section 3 along with its theoretical properties and its simple algorithm. In Section 4, we conduct experiments on simulated and real data to assess the quality, robustness, and time complexity of UMFI compared to MCI. Finally, an overview of the work, its limitations, and ideas for future work are discussed in Section 5.

**Related work**

This paper is greatly inspired by the development of marginal contribution feature importance (MCI) by Catav et al. [11]. Although other methods, such as SAGE [14], have been retooled to better explain data [12], up until this point, MCI had been the only feature importance method developed specifically to explain data. Let $F = \{x_1, ..., x_p\}$ be the set of features used to predict the response variable, $Y$. Recall that the universal predictive power of a set of features $S \subseteq F$ is given by

$$\nu(S) = \min_{f \in G(\emptyset)} \mathbb{E}[l(f(\emptyset), Y)] - \min_{f \in G(S)} \mathbb{E}[l(f(S), Y)], \tag{1}$$

where $l$ is a specified loss function and $G(S)$ is the set of all predictive models restricted to using features in $S \subseteq F$. $\nu$ is closely related to mutual information, with equality under ideal conditions [14], and in practice, $\nu$ is often approximated by machine learning evaluation functions. Using this, Catav et al. [11] defined the marginal contribution feature importance (MCI) of a feature $x_i \in F$ by

$$I_\nu(x_i) = \max_{S \subseteq F} \nu(S \cup \{x_i\}) - \nu(S). \tag{2}$$

To achieve our goal of improving upon the shortcomings of MCI, we evaluate the importance of a feature of interest $x_i$ after preprocessing the data to remove dependencies on $x_i$. Finding independent representations of predictors for creating improved feature importance methods is a novel objective, though similar ideas have been suggested as future work in König et al. [30] and Chen et al. [12]. The weaker concept of finding orthogonal representations of data has been discussed previously [18], though the discussion has been limited to relative importances measures for multiple linear regression, mostly in the domain of psychology [6, 52]. While orthogonalizing predictors can be done easily with simple techniques, methods which can not only remove correlations between features, but also remove more general dependencies, have seen great progress within the domains of AI fairness and privacy. Some examples of these techniques include regression [7], optimal transport [28], neural networks [10, 41], convex optimization [10], and principal inertial components [45]. Linear regression and optimal transport were implemented for UMFI in this paper.

## 2  Axioms for explaining data

Any attempt to build a method that explains the data should begin by rigorously defining what explaining the data truly means. Different definitions and goals have been formulated by Chen et al. [12] and Catav et al. [11]. Inspired by these definitions, we provide three intuitive, justified, and rigorous axioms for true-to-data feature importance methods. Given a feature set $F$, a response $Y$, and a feature of interest $x_i \in F$, the feature importance of $x_i$ is defined as $Imp^{F,Y}(x_i) \in \mathbb{R}_{\geq 0}$. We define the following three axioms as vital for any method that claims to explain the data:

1. **Elimination axiom:** Eliminating a feature $x_j$ from the feature set $F$ can only decrease the importance of the feature of interest:

$$\forall x_i \in F \setminus \{x_j\}, Imp^{F \setminus \{x_j\}, Y}(x_i) \leq Imp^{F,Y}(x_i).$$

2. **Duplication invariance and symmetry axiom:** Adding a duplicate copy of a feature $\hat{x} = x_j$ already in the feature set $F$ will not change the importance of the other features in $F$, and the duplicated feature will have importance equal to the original feature:

$$\forall x_i \in F, \ Imp^{F,Y}(x_i) = Imp^{F \cup \{\hat{x}\},Y}(x_i) \text{ and } Imp^{F \cup \{\hat{x}\},Y}(\hat{x}) = Imp^{F \cup \{\hat{x}\},Y}(x_j).$$

3. **Blood relation axiom:** If data is generated from a causal graph, feature $x_i$ will be given non-zero and positive importance if and only if it is blood related to the response $Y$ in the causal graph. Two vertices in a causal graph are said to be blood related if there is a directed path between them or if there is a backdoor path between them via a common ancestor.

$$Imp^{F,Y}(x_i) > 0 \iff x_i \in BR(Y).$$

The elimination axiom comes directly from Catav et al. [11]. Once a feature is observed to be significantly related to the response, the relationship strength between the feature and response should not drop, regardless of the additional features added. In fact, often times the importance should increase since adding features could reveal further synergistic information about the response $Y$.

The duplication invariance and symmetry axiom separates feature importance methods that are for data explanation from methods intended for model optimization [11]. A model may use the two identical features equally often and therefore spread the importance equally between them (random forests), or only one of the features may be given importance (lasso) [12]. However, from the data's perspective, both features should be equally related to the response and the original importance found before duplication should still be true. Further, after duplication, no additional interaction capability is available [22], so the importance of all other features should remain the same.

The blood relation axiom asserts that feature importance scores intended for data explanation should extract reliable knowledge about the underlying causal graph and data generating process. A statistical association between a feature and the response, which is a quality of interest for many applications (e.g., genome-wide association studies), exists precisely when the two features are blood related, or equivalently, when there is an open path between them (see Greenland et al. [20] and Williams et al. [48] for a more in-depth explanation of this definition as well as other relevant concepts about causal graphs). Thus, a feature importance metric satisfying this axiom would give non-zero importance to a feature if and only if there is a statistical association between that feature and the response. Additionally, if the goal is to construct a causal graph to represent the relationships in the data, then a feature importance metric satisfying this axiom can partition the feature set into features that are blood related to the response and features that are not blood related to the response. Although it does not enable us to immediately recover the full causal graph, this partitioning may be a helpful supplemental tool for other causal discovery methods. See Supplement B for further discussion.

## 3 Ultra-marginal feature importance

Let $F = \{x_1, ..., x_p\}$ be a set of $p$ features of arbitrary type used to predict the response $Y$. We note that features may be viewed as random variables, or as realizations of random variables according to their joint distribution, in the form of a dataset.

In order to define ultra-marginal feature importance, we require that the evaluation function $\nu$, which measures the predictive power of a group of features [11], and which approximates Equation (1), is also defined for transformations of the feature set following the removal of dependencies. We therefore define the space of information subsets of a feature set $F$ as $\mathcal{I}(F) = \{g(F) : g \text{ is any function defined on } F\}$. We call these information subsets of $F$ because $I(Y; g(F)) \leq I(Y; F)$ holds for any function $g$ by Theorem A.3.

**Definition 1.** *We denote $S_{x_i}^F$ as a preprocessed feature set after dependencies on the feature of interest $x_i$ have been removed from $F$. An optimally preprocessed feature set is denoted by $\hat{S}_{x_i}^F$, and we say that a preprocessing $S_{x_i}^F$ is optimal if it obeys the following properties:*

*1. $S_{x_i}^F = g(F)$ for some function $g$*

*2. $S_{x_i}^F \perp\!\!\!\perp x_i$*

*3. $I(Y; S_{x_i}^F, x_i) = I(Y; F)$*

The first property ensures that $S_{x_i}^F \in \mathcal{I}(F)$, and hence, no information from outside of $F$ is gained during the transformation. The second property upholds that the random vector $S_{x_i}^F$ is independent of $x_i$, and the last property affirms the optimality of $S_{x_i}^F$ in the sense that there is no unnecessary information loss incurred during preprocessing. Given that it exists, an optimal preprocessing $\hat{S}_{x_i}^F$ is not unique, since scaling $g(F)$ by a constant does not affect the last two properties. In practice, the last two properties can be difficult to guarantee, but we see later in Section 4 that non-optimal preprocessings are good enough in many circumstances.

**Definition 2.** *Given an evaluation function $\nu : \mathcal{I}(F) \rightarrow \mathbb{R}_{\geq 0}$ and a feature set $F$, we define the ultra-marginal feature importance (UMFI) of a feature $x_i \in F$ as*

$$U_\nu^{F,Y}(x_i) = \nu(S_{x_i}^F \cup \{x_i\}) - \nu(S_{x_i}^F). \tag{3}$$

UMFI obeys the three axioms given in Section 2 under certain assumptions as proven in Appendix C. Mainly, we assume that $\nu(\cdot) \approx I(Y; \cdot)$. Under ideal conditions, this relationship holds when $\nu$ satisfies Equation (1) [14], but in practice, the accuracy of the approximation depends on the quality of the method, the specified loss function, and the response variable's distribution [15]. See Covert et al. [15] and Appendix A.3 for a more thorough overview.

Since UMFI is model-agnostic, we provide a general algorithm for computing the ultra-marginal feature importance of a feature $x_i \in F$, which can be applied using any pair of preprocessing and modeling techniques. We note that $\nu_f$ is not restricted to the domain of machine learning models or even models in general. For example, one could also implement UMFI with measures of dependence such as the Hilbert–Schmidt independence criterion [21] or non-ML estimates of mutual information [31]. Furthermore, if machine learning modeling techniques are used for UMFI, we advise that the median score over multiple iterations of the algorithm is used to account for the variance of $\nu_f$.

---

**Algorithm 1:** Algorithm for computing UMFI

1: Let $Y$ be the response variable of the set of predictors $F$. Choose a feature $x_i \in F$.
2: Obtain $S_{x_i}^F$ by using a technique that optimally removes dependencies on $x_i$ from $F$.
3: Specify a method $f$ and a corresponding evaluation function $\nu_f$.
4: Estimate the predictive power, $\nu_f(S_{x_i}^F)$, that $S_{x_i}^F$ has about $Y$.
5: Estimate the predictive power, $\nu_f(S_{x_i}^F \cup \{x_i\})$, that $S_{x_i}^F \cup \{x_i\}$ has about $Y$.
6: **return** $U_{\nu_f}^{F,Y}(x_i) = \nu_f(S_{x_i}^F \cup \{x_i\}) - \nu_f(S_{x_i}^F)$

---

# 4 Experiments

We perform experiments to compare UMFI and MCI with respect to quality, robustness, and time complexity. To implement UMFI, we consider optimal transport [28] (UMFI_OT) and linear regression [7] (UMFI_LR) as methods to remove dependencies from the data. A detailed overview of these implementations is shown in Appendix E and experiments comparing these methods appear in Appendix F. For all experiments, we use random forests' out-of-bag accuracy ($R^2$ OOB-accuracy for regression tasks and OOB classification accuracy for classification tasks) as the evaluation metric $\nu_f$ [8]. We use the *ranger* R package to implement random forests with default hyperparameters and 100 for the number of trees [50]. All experiments were run in Microsoft R Open Version 4.0.2 [35]. Appendix G contains additional experiments comparing UMFI and MCI with other feature importance metrics including ablation, permutation importance, and conditional permutation importance. In the same section, we rerun the experiments comparing MCI and UMFI using extremely randomized trees instead of random forests and do an additional comparison on a real dataset from hydrology [1]. Code for all experiments can be found in the Supplement.

## 4.1 Experiments on simulated data

We run UMFI on simulated data to verify that it performs well compared to MCI. The data in all simulation studies contains one response variable $Y$, four explanatory features $x_1, x_2, x_3, x_4$, and 1000 randomly generated observations. Each study is repeated 100 times to test stability.

### 4.1.1 Nonlinear interactions

Interaction effects are common in many scientific disciplines where assessing feature importance is prevalent, including hydrology [27, 2, 32], genomics [11, 47, 37], and glaciology [17, 4, 9, 39]. So, as was done in Catav et al. [11], we assess the ability of MCI and UMFI to detect nonlinear interaction effects in the data [34]. We consider:

$$x_1, x_2, x_3, x_4 \sim \mathcal{N}(0, 1)$$
$$Y = x_1 + x_2 + sign(x_1 * x_2) + x_3 + x_4.$$

Feature importance metrics should ideally conclude that $x_1$ and $x_2$ have higher importance compared to $x_3$ and $x_4$ because of the extra interaction term, $sign(x_1 * x_2)$. Figure 1a shows consistently good performance across all methods. Each method gave high relative importance scores to $x_1$ and $x_2$, while $x_3$ and $x_4$ received less, but still substantial importance. All methods show similar variability.

### 4.1.2 Correlated interactions

Interacting features are often correlated [25, 27]. So, this simulation study aims to repeat the nonlinear interactions study, except now $x_1$ and $x_2$ are highly correlated with eachother. In the same way, $x_3$ and $x_4$ are highly correlated with eachother. Let $A, B, C, D, E, G \sim \mathcal{N}(0, 1)$. We consider:

$$x_1 = A + B, \ x_2 = B + C, \ x_3 = D + E, \ x_4 = E + G$$
$$Y = x_1 + x_2 + sign(x_1 * x_2) + x_3 + x_4.$$

Just as with the interaction experiment with independent features, we would expect $x_1$ and $x_2$ to be more important than $x_3$ and $x_4$ because of the extra interaction term, $sign(x_1 * x_2)$. The results in Figure 1b clearly show that UMFI provides better estimations of feature importance compared to MCI when correlated interactions are present. MCI estimates that all features have approximately the same feature importance scores, while both UMFI methods show significantly greater importance for $x_1$ and $x_2$ compared to $x_3$ and $x_4$. MCI fails in this experiment because it penalizes feature subsets that share information with the feature of interest $x_i$ when evaluating the importance of $x_i$ via Equation (2). For example, if we are assessing the MCI score for $x_1$, since $x_2$ is strongly correlated with $x_1$, then the predictive power offered by $x_1$ on top of a subset $S$ would be diminished by the presence of $x_2 \in S$. Therefore, $x_2$ is not utilized in the MCI score for $x_1$, which prevents the detection of the interaction term $sign(x_1 * x_2)$. UMFI is able to detect this interaction because it can extract the information from $x_2$ that interacts with $x_1$ while keeping this extracted feature independent of $x_1$. Although not yet tested, we suspect that similar results would be demonstrated in the presence of dependent, but uncorrelated interactions.

### 4.1.3 Correlation

Feature importance methods that seek to explain data, such as MCI and UMFI, should not change the measured importance of features in the presence of highly correlated or duplicated variables according to the duplication invariance and symmetry axiom. To test this, we implement a simulation study similar to the ones found in Catav et al. [11]. Let $\epsilon \sim \mathcal{N}(0, 0.01)$. We consider:

$$x_1, x_2, x_4 \sim \mathcal{N}(0, 1), \ x_3 = x_1 + \epsilon$$
$$Y = x_1 + x_2.$$

The addition of $x_3$, which is approximately a duplicate of $x_1$, should not alter the importance of $x_1$, and $x_1$ should remain equally as important as $x_2$, since they have the same influence on the response $Y$. The results shown in Figure 1c show that both MCI and UMFI work reasonably well. As with the previous simulation experiment, the variability is consistent across methods. As was desired, UMFI with linear regression shows equal relative importance scores for $x_1$ and $x_2$. The importance given to $x_2$ was slightly greater than $x_1$ according to MCI and UMFI with optimal transport. Interestingly, MCI assigns some importance to $x_4$, which was independent of the response, while both UMFI methods assign importance scores close to zero. Because of this, we conclude that UMFI with linear regression performs the best in this simulated scenario.

### 4.1.4 Blood relation

To ensure that UMFI is true to the data and could be used to learn part of the structure of the causal graph in theory as well as in practice, we implement the blood relation simulation experiment. In this

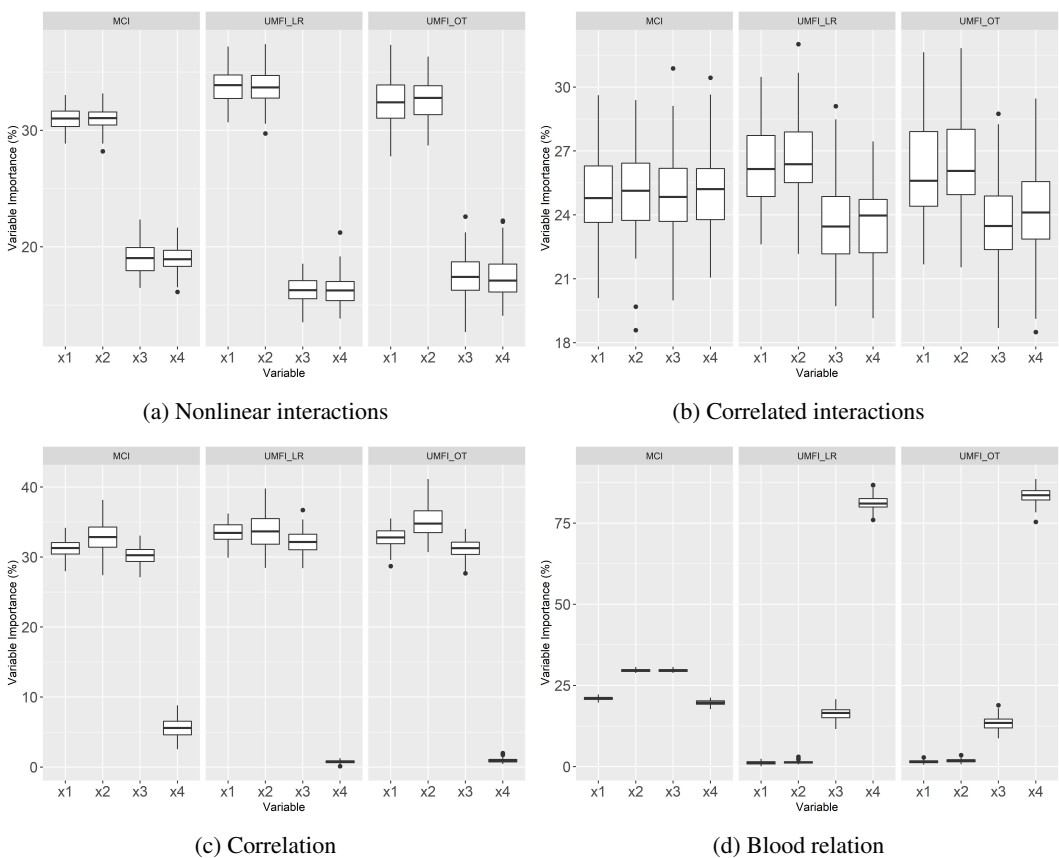

(a) Nonlinear interactions           (b) Correlated interactions

(c) Correlation           (d) Blood relation

Figure 1: Results for the experiments on simulated data from Subsection 4.1. Feature importance scores are shown as a percentage of the total for each of $x_1$ to $x_4$ from 100 replications. Results are shown for marginal contribution feature importance (MCI), ultra-marginal feature importance with linear regression (UMFI_LR), and ultra-marginal feature importance with pairwise optimal transport (UMFI_OT).

study, data is generated from the causal graph in Figure 7 from the Supplement, which was inspired by the collider causal graph found in Harel et al. [23]. The feature $S$ is unobserved, thus only $x_3$ and $x_4$ are blood related to the response $Y$. Because of this, according to the blood relation axiom, $x_3$ and $x_4$ should be given high and positive importance while $x_1$ and $x_2$ should receive zero importance. In Section 3, we proved that in ideal scenarios, UMFI will only give non-zero importance to blood related features. We hypothesize that we can extend this to real-world scenarios where non-Gaussian features and interaction information appear. To test this, we consider:

$$x_1, S \sim \mathcal{N}(0,1), \; \delta \sim \mathcal{U}(-1,1), \; \epsilon \sim \mathcal{U}(-0.5, 0.5), \; \gamma \sim Exp(1)$$
$$x_2 = 3 * x_1 + \delta, \; x_3 = x_2 + S$$
$$Y = S + \epsilon$$
$$x_4 = Y + \gamma.$$

The results shown in Figure 1d indicate that MCI fails to distinguish the blood related features, since most of the importance is given to $x_1, x_2 \notin BR(Y)$. In contrast, UMFI_LR and UMFI_OT detect that $x_1$ and $x_2$ should have zero importance while giving most of the importance to $x_4$ and the rest of the relative importance to $x_3$.

## 4.2 BRCA experiments

We use the same breast cancer (BRCA) classification dataset [43] used in previous feature importance studies including Catav et al. [11] and Covert et al. [14] to test the quality and robustness of UMFI

on real data. The original data contains over $17,000$ genes and $571$ anonymous patients that have been diagnosed with one of $4$ breast cancer sub-types. We consider the same subset of $50$ genes as in Catav et al. [11] and Covert et al. [14] for easier computation and result visualization. Of the $50$ selected genes, $10$ are known to be associated with breast cancer, while the other $40$ genes are randomly sampled. This data was downloaded from `https://github.com/TAU-MLwell/Marginal-Contribution-Feature-Importance/tree/main/BRCA_dataset` (MIT License). In Catav et al. [11] and Covert et al. [14], these $40$ randomly sampled genes are assumed to be unassociated with breast cancer. However, to ensure a more definitive ground truth, we also randomly permute the values of these $40$ genes across their respective $571$ observations to further reduce the chance that these genes have any association with breast cancer. Quality is then measured with the true positive and true negative rates: the $10$ BRCA associated genes should have some non-zero importance (positive), and the other $40$ genes should have exactly zero importance (negative). These experiments were run $200$ times on different seeds and with a different random sample of $500$ patients for each iteration. Robustness is measured using the standardized interquartile range (SIQR) from the repeated experiments, which is calculated by dividing the average IQR across the $50$ features by the average median. This experiment is too computationally intensive for MCI to be calculated exactly, so we implement MCI assuming soft 2-size submodularity.

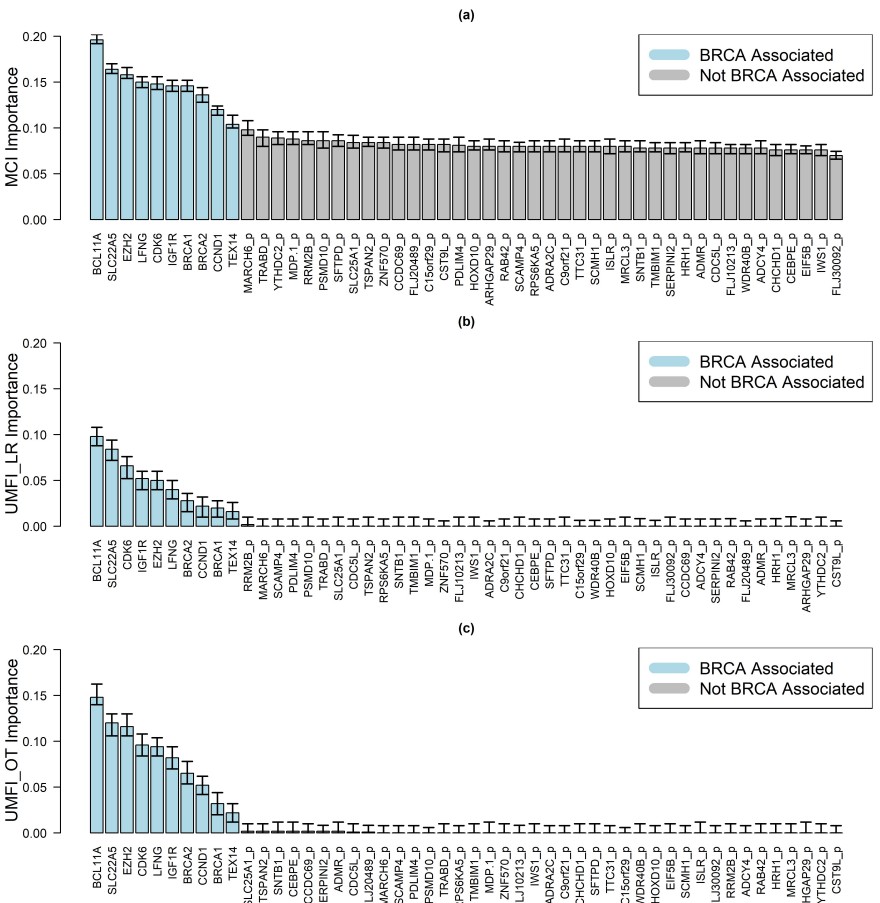

Figure 2: Median feature importance scores provided by (a) MCI, (b) UMFI with linear regression, and (c) UMFI with pairwise optimal transport, for each gene in the BRCA dataset after 200 iterations. Genes colored in blue are known to be associated with breast cancer while genes colored in grey are random permutations of randomly selected genes, which we assume to be unassociated with breast cancer. The first and third quantiles of the scores are visualized for each gene.

We found that MCI and UMFI (UMFI_LR and UMFI_OT) correctly gave significant importance to the $10$ genes that are known to be associated with breast cancer (Figure 2). Interestingly, the ordering of important features was similar across methods, with BCL11A and SLC22A5 always

being the most important and TEX14 always being the least important of the 10 BRCA-associated genes. However, MCI consistently gives non-zero importance to all features, while UMFI correctly gives zero importance to the majority of the randomized genes. Furthermore, UMFI's performance in this experiment improves with increased iterations. After running the experiment 5000 times, both UMFI methods have a perfect overall accuracy when distinguishing between important and permuted features (Appendix G.2.1). Although UMFI scores have higher variability than MCI (Table 1), it is clear from Figure 2 that UMFI separates the 10 associated genes from the 40 unassociated genes better than MCI does.

Table 1: The standardized interquartile range (SIQR), true positive rate (TPR), true negative rate (TNR), overall accuracy (OA), and the number of features for which feature importance can be calculated within 1, 15, and 60 minute(s) are displayed after running the methods on the BRCA data.

| Method | SIQR | TPR | TNR | OA | @1min | @15min | @1hr |
|---|---|---|---|---|---|---|---|
| MCI (k=2) | **6.6 %** | **1** | 0 | 0.20 | 35 | 80 | 130 |
| UMFI (LR) | 41.9% | **1** | **0.975** | **0.98** | **500** | **2000** | **4010** |
| UMFI (OT) | 28.5% | **1** | 0.775 | 0.82 | 300 | 1500 | 3000 |

## 4.3 Computational complexity

MCI must train and evaluate a model for each element of the power set of the feature set, which implies $O(2^p)$ model trainings if there are $p$ features. If the evaluation function $\nu$ obeys soft $k$-size submodularity, then the maximizing subset has no more than $k$ elements, which reduces the number of model trainings to $O(p^{k+1})$ [11]. UMFI circumvents the exponential training time since it can be evaluated immediately after removing the dependencies of $x_i$ from the feature set $F$. To confirm the above statements, and to show that the extra model trainings required for MCI dominate the computation time for removing dependencies in UMFI, we ran a simple experiment. For a range of dataset sizes from the BRCA data, we evaluate the computation time for calculating the feature importance scores of all features using MCI and UMFI. We ran this experiment for a dataset with 5 features, and then slowly added features until our given time budget of 1 hour ran out. Once all 50 BRCA features were used, more features were randomly generated. All datasets had 571 observations. These experiments were run using an Intel Core i9-9980HK CPU 2.40GHz with 32GB of RAM. Code was parallelized in R, and 12 of the 16 available threads were used.

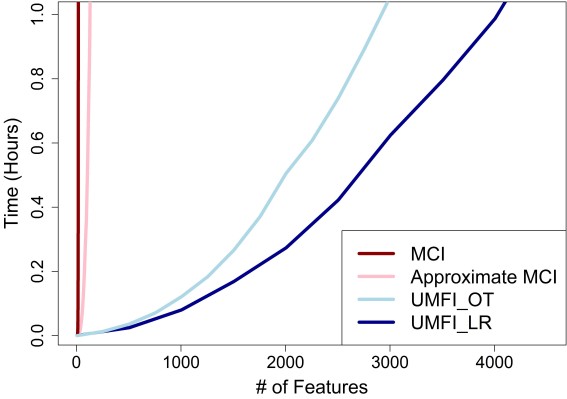

Figure 3: Computation time for a single iteration of each method including: MCI (dark red), MCI with the soft 2-size-submodularity assumption (pink), UMFI_OT (light blue), and UMFI_LR (dark blue), plotted against the number of processed features from the BRCA data.

From Figure 3, we can observe that UMFI is approximately superlinear, with UMFI_OT incuring more computational cost compared to UMFI_LR. Giving each method 1 hour to run, MCI processed

19 features, MCI with the soft 2-size submodularity assumption processed 130 features, UMFI_OT processed about 3000 features, and UMFI_LR processed about 4000 features (Table 1).

## 5   Conclusion

In this study, we introduced ultra-marginal feature importance (UMFI), a new method that uses preprocessing techniques, originally developed in the domain of AI fairness, to provide fast and accurate feature importance scores for the purposes of explaining data. We introduced three ideal axioms that feature importance measures should satisfy if they claim to explain the data, which are all satisfied by UMFI under some basic assumptions (Appendix C). Optimal transport and linear regression were explored as preprocessing techniques to remove dependencies from data. When compared with MCI, the previous state-of-the-art method for explaining data, experimental results showed that UMFI was able to provide faster and more accurate estimates of feature importance on real and simulated data, particularly in the presence of correlated interactions and unrelated features. UMFI's superior time complexity could be leveraged to run feature importance on larger datasets or to achieve more accurate results by utilizing its median scores after many iterations.

Throughout the work on this paper, several shortcomings appeared. First, we only considered two simple methods for removing dependencies, linear regression and pairwise optimal transport. Other methods certainly exist in the literature, including optimal transport with chaining [28], neural networks [10, 41], or principal inertial components [46]. Though our two methods performed fairly well on the real and simulated datasets in Section 4, optimal transport and linear regression failed to find representations of the data that were independent of the protected attribute when we tested the methods on a hydrology dataset with more shared information compared to BRCA [1] (Appendix G.4). However, neural nets or principal inertial components certainly could have given better results. Also, despite requiring significantly more computational cost, better methods for estimating the conditional CDF, or using optimal transport with chaining, should give better estimates for $S_{x_i}^F$ when implementing UMFI_OT. Even though dependencies were not removed optimally for the hydrology dataset, the estimates of feature importance were still reasonably accurate. Second, UMFI scores are less robust than MCI since they have higher variability, however, because of the significantly lower computational cost, UMFI can be run multiple times and averaged to increase robustness. Third, it is not clear how closely $\nu_f$ approximates mutual information in practice. Finally, though UMFI can work for any arbitrary feature type, in this paper, we have only considered datasets with continuous explanatory variables.

In future work, we would like to test how well other methods, such as neural networks, pair with UMFI while further testing on a wider variety of random variable types such as binary, categorical, and ordinal features. Further, we would like to explore how well dependence can be removed and UMFI can be estimated on real data as the number of features increases to sizes much larger than 50.

To reiterate, UMFI is a powerful tool for detecting and explaining the relationships hidden within data. We emphasise that UMFI is just a framework. A variety of other methods can be used to estimate the universal predictive power $\nu$ including, but not limited to, XGBoost, neural networks, or Gaussian processes. Even non-model-based methods such as Hilbert-Schmidt independence criterion could be explored in future applications. Furthermore, new preprocessing techniques for dependence removal are still being developed in the AI fairness community, so these, in addition to other existing methods, can be used in future applications of UMFI for additional improvements.

## Broader Impact

We hope that UMFI will be a useful tool in a variety of disciplines including bioinfomatics, ecology, earth sciences, and health science for discovering scientific processes and relationships hidden within data. Though we think that our contributions can only lead to positive social and environmental impacts by aiding scientific discoveries in domains like earth science and bioinformatics, statistical methods, especially those that are aimed at genetics research, have historically been used to justify harmful and misleading claims. If such claims arise using our methods, then they should be dismissed since direct causal effects cannot be concluded after using our methods alone.

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
