# OpenReview forum: "Ultra-marginal Feature Importance"
_NeurIPS.cc/2022/Conference — NeurIPS 2022 Submitted_

### Official Review · Reviewer_qdSP · 2022-07-08

**Rating:** 8
**Confidence:** 4
**Soundness:** 4 excellent
**Presentation:** 4 excellent
**Contribution:** 4 excellent

**Summary:**

The problem of estimating feature importance is considered, specifically
estimating marginal contribution importance (MCI). In essence, MCI estimates the
value of a feature based on the performance drop when the feature is excluded.
This paper studies an extension which the authors dub ultra-marginal feature
importance (UMFI) that maximises over "information subsets" rather than the all
subsets of features (like the original MCI).

It turns out this slight change provides large computational benefits and
improved performance to boot. The authors prove a key result that allows the
maximising information subset to be approximated using dependency removal
techniques such as those from the fair learning literature. This reduction in
complexity allows scaling to previously unobtainable number of features.


**Questions:**

It's clear many methods could be used as ν, and is discussed in the conclusion, but
no alternative is explored in the empirical results. It's quite natural to wonder how
critical the choice of supervised learning methods is: have this been explored?

The choice of ν can also be more generic dependency measures, so what about
easy to estimate measures of dependence like HSIC?


**Limitations:**

It's clear many methods could be used as ν, and is discussed in the conclusion, but
no alternative is explored in the empirical results. It's quite natural to wonder how
critical the choice of supervised learning methods is: have this been explored?

The choice of ν can also be more generic dependency measures, so what about easy
to estimate measures of dependence like HSIC? Exploration of supermodularity
might be more straightforward.


**Strengths And Weaknesses:**

This is a great extension to the MCI framework as it is simple, novel, and has a
large practical impact. The presentation is excellent, theorem proof
straightforward, and empirical experiments well chosen. Limitiations of the
study are well discussed.

One weakness is the authors have found a dataset where their choice of optimal
transport/linear regression for dependence removal failed, but it is relegated
to supplementary materials.

minor: line 301 is missing the word "features"

---

> ### Author Response · Authors · 2022-08-01
> **Response to Reviewer qdSP**
>
> Reply #1: We thank reviewer qdSP for the kind review, and more importantly for pointing out the HSIC method, which we were unfamiliar with. Please see the updated version of our paper as well as our more specific responses below.
>
> **Reviewer: One weakness is the authors have found a dataset where their choice of optimal transport/linear regression for dependence removal failed, but it is relegated to supplementary materials.**
>
> Reply #2: Indeed we put the hydrology experiment in the supplement, but we do point out this limitation in the conclusion of the main text. We put the hydrology experiment in the supplement because of space constraints, and we wanted to be consistent with the previous feature importance papers which used BRCA as their main “real data” experiment. We would like to emphasize that UMFI is just a framework. An innumerable amount of different methods are available for removing dependencies, but we just choose two simple ones to demonstrate the potential capabilities of our framework. Even when the removal of dependencies did not occur optimally, as was the case in the hydrology example, the feature importance outputs were still reasonable, which we find encouraging. We emphasize that the reasonable results in hydrology in the face of nonoptimal dependency removal are encouraging in the revised version of the conclusion (Lines 282-291).
>
> **Reviewer: minor: line 301 is missing the word "features"**
>
> Reply #3: Thanks! This is fixed in the new version.
>
> **Reviewer: It's clear many methods could be used as $\nu$, and is discussed in the conclusion, but no alternative is explored in the empirical results. It's quite natural to wonder how critical the choice of supervised learning methods is: have this been explored?**
>
> Reply #4: Yes, many methods can be used to calculate $\nu$, and we think this is one of the beautiful properties of UMFI and MCI. In the original supplement, we ran the same simulation studies comparing MCI and UMFI with extremely randomized trees instead of random forests. We found that the main conclusions of these simulation results do not change for random forests vs extremely randomized trees. The hydrology example was run with extremely randomized trees instead of random forests as well. All of these experiments are also in the revised text.
>
> **Reviewer: The choice of $\nu$ can also be more generic dependency measures, so what about easy to estimate measures of dependence like HSIC?**
>
> Reply #5: We had not heard of HSIC before you pointed this out. Thank you for informing us of this method as we had previously wondered if such a method existed. We add this method in our discussion of applicable methods for UMFI (Lines 141-142) and may implement it for future work (Lines 303-304).

---

### Official Review · Reviewer_sCQt · 2022-07-11

**Rating:** 7
**Confidence:** 4
**Soundness:** 3 good
**Presentation:** 3 good
**Contribution:** 3 good

**Summary:**

This work considers how to quantify the importance of different features in a ML model (for each feature $f$ from the set of all features $F$) to the response variable $Y$. To do so, they modify a method known as "marginal contribution feature importance" (MCFI). In simple terms, rather than measuring the maximum contribution that a feature makes to a subset of other features (where contribution roughly means how much it improves predictive performance), "ultra marginal feature importance" (UMFI) measures the maximum contribution to any random variable that is a function of the full feature set.

This sounds like it could make computation harder, but in fact the authors prove that it can be computed efficiently by removing $f$'s dependencies/signal from $F$ (in a sense that isn't precisely defined, in my view), and then measuring how the newly trained model's performance degrades. Thus, the computational cost is significantly lower than computing the maximum contribution over $2^{d-1}$ feature subsets (where there are a total of $d$ features).

The authors' proof rests on several assumptions that won't hold in practice, but the method should still give reasonable results. And there is no gold-standard way of removing dependencies, but the authors demonstrate the method with a couple viable options (e.g., residualizing out with linear regression).

**Questions:**

On line 120, is it important that $g$'s domain is $A \subseteq F$ rather than simply $F$? I don't see why it can't simply be $F$. And is it important that the co-domain be $\mathbb{R}^{|A|}$ rather than some higher- or lower-dimensional Euclidean space?

A couple points from above that would be helpful to comment on:
- Non-uniqueness of dependency-removing transformation
- How MCI handles correlated features wrong, and how UMFI is supposed to fix the problem
- Why MCI gives non-zero importance to unrelated features but UMFI somehow does not
- Various points about the experiments

**Limitations:**

The authors included a nice discussion of limitations towards the end of the paper. One additional point that I would have liked to see covered, besides the main challenge of properly removing dependencies: whether there are any downsides to finding the maximum contribution. For example, are there cases where finding the max would yield equal importance for two features $x_1$ and $x_2$, where $x_1$ is useful on its own but $x_2$ is only useful when present along with a complementary feature $x_3$? This seems like a potential issue for both MCI and UMFI (depending on what one wants out of their importance values).

**Strengths And Weaknesses:**

### Strengths

- This work presents a new perspective on how to define feature importance in ML models. Rather than removing/corrupting subsets of features, it considers the maximum contribution that each feature $f$ makes to any random variable that's a function of the full feature set. Leave-one-out or Shapley value approaches wouldn't make sense in the context of arbitrary functions of the features (the value function $\nu$ is no longer a set function, but a function of any random variable), but it's a neat result that the random variable to which a feature $f$ contributes most is somewhat easy to approximate.
- UMFI seems to provide reasonable results in the experiments.

### Weaknesses

About the method:
- The idea of removing dependencies of one random variable from another does not seem to be precisely defined anywhere in the paper. It seems like there is not a unique way to do this - for example, I can remove $f_1$'s dependencies from $f_2$ by setting $f_2$ to a constant. If the only requirement in this work is that the modified $f_2$ variable becomes independent from $f_2$, and the non-uniqueness of the transformation isn't a problem, that should be discussed more prominently.
- The name "information subsets" for $I(F)$ was a bit confusing, it doesn't seem like there are subsets involved here (the way they are in MCI or SAGE). It seems to mean the set of all possible functions applied to the features/input variables. I guess I get the spirit of the name, but it's confusing.
- On lines 55-57, one of the stated issues with MCI is that it underestimates the importance of correlated features. I didn't quite understand how MCI gets correlated features wrong, or how UMFI is supposed to fix the problem - can you elaborate on this? This point doesn't seem very well supported in theory, and it's only supported in a limited sense by the experiments.
- On lines 57-58, one of the stated issues with MCI is that it can give non-zero importance to features unrelated to the model. As UMFI yields strictly larger importance than MCI (this is mentioned later in the paper), UMFI should share this flaw rather than fixing it. The results in Figure 2 don't seem to reflect this, which is very strange.
- I believe the authors are potentially confusing two different meanings of the term "marginal." The authors labeled both MCI and UMFI as "marginal methods", but I don't think they are if you look at what the "marginal vs. conditional divide" is about. The divide is about whether a feature importance method handles held-out features with their marginal or conditional distribution; e.g., KernelSHAP often uses the marginal distribution, and SAGE advocates for using the conditional. Because MCI retrains models for each subset of features, it's actually a *conditional* method, at least approximately (if it's helpful, I can point to a paper that proves how retraining and sampling held-out features from their conditional distribution are approximately equivalent). UMFI on the other hand is *neither* - the models are trained on completely different features so there's no question of re-using a model and sampling replacement values for held-out features. For MCI, "marginal" refers to "marginal contribution" from the game theory context, which has nothing to do with the marginal vs. conditional label. UMFI is also roughly about marginal contributions (even though $\nu$) isn't a cooperative game, so I could see calling it a "marginal contribution" method. But calling both of these "marginal methods" isn't very helpful given the ambiguity.

Some potential issues with the experiments:
- The experiments don't seem to use any real datasets. Even the BRCA experiment synthetically modifies certain features to ensure that they're unimportant to the response variable. Would it be possible to use a real dataset, or at least present additional results with the unmodified BRCA data?
- Related to the request for a real dataset, another type of metric would be to measure the predictive performance of models trained with the most important features - and this would be simple to run even with real datasets. Would the authors consider adding something like this?
- Would it be possible to include other baselines in the experiments? For example, the methods examined in MCI or SAGE?
- MCI giving large importance to randomized features in Figure 2a is a bit hard to believe, it's very counterintuitive. Can you verify that there isn't a mistake here, or explain why this occurs?

The notation in this work was confusing at times. A couple choices that made this paper difficult to read were:
- $f$ denotes a function in most ML papers, but here it denotes a feature. Many papers instead represent the features as $x = (x_1, \ldots, x_d)$ when there are $d$ features, and the model inputs/outputs are usually written as $(x, y)$. $y$ is retained here, but not $x$ for some reason
- It would be helpful to write explicitly that $F$ is simultaneously used as both a set (e.g., $S \subseteq F$) and a random variable (e.g., $I(Y; F)$). Similar to the above, the notation that's often used is $S \subseteq D = ${$1, \ldots, d$} to denote the feature indices and $x_S$ to represent a random variable that concatenates the set of features
- It would be helpful to say explicitly that $g(A)$ represents a random variable resulting from applying a function $g$ to a subset of features (towards the beginning of section 3)
- The symbol used to denote independence (see theorem 3.1) is never defined and likely unfamiliar to some readers
- $g$ is overloaded to denote two different types of functions: one is a predictive model for $Y$ (equation 2) and one is an arbitrary function on the feature space (definition of $I(F)$)

Nits:
- On line 49, there are methods that suggest training models with many feature subsets (another one is SPVIM, Williamson & Feng 2020), but several do not - including SAGE and permutation tests. There are a variety of tricks for handling the held-out features including setting to the mean, sampling replacement values (from the dataset or from a generative model), training a model to handle missing features, etc (see "Explaining by removing: a unified framework for model explanation" by Covert et al.)
- On lines 29-30, marginal (rather than conditional) methods are not typically thought of as being for interpreting the data. See "True to the model or true to the data" by Chen et al., it discusses how marginal methods are better suited for understanding the model rather than the data
- I think there's a typo in the subscript of the union operator on line 134 - shouldn't it be $f$ rather than $F$?

---

> ### Author Response · Authors · 2022-08-01
> **Response to Reviewer sCQt: Part 1**
>
> Reply #1: We thank reviewer sCQt for their response. It is clear that the reviewer spent a great deal of time trying to understand our paper thoroughly and providing many useful suggestions. We are especially grateful for their suggestion to make our theoretical points more rigorous. Please see our revised paper and specific responses to the points raised below. We look forward to discussing these topics further with you.
>
> **Reviewer: Weaknesses about the method**
>
> Reply #2: In the revised work, we defined what it means to optimally remove dependencies with three criteria in Definition 2. Still, the optimal preprocessed feature set is not necessarily unique, thus we added an explicit sentence about the non-uniqueness of $S^F_x$ in the updated text (Lines 127-128).
>
> Reply #3: We understand your concern, but over time, we have grown quite fond of the term “information subsets”, but if you have any better suggestions for a name that is intuitive/concise/clear, we would be willing to change it.
>
> Reply #4: It is not correlated features that necessarily cause issues in MCI, but rather correlated features that carry synergistic information about the response (e.g., $cor(x_1,x_2)=0.8$ and $Y=x_1+x_1*x_2+x_3$). In essence, when MCI is calculated for $x_1$, if $x_2$ shares a lot of information with $x_1$ then it cannot usually be included in the subset that maximizes the difference in the evaluation function, **even if the two features contain synergistic information (interaction effects) about the response**. We further clarified this in lines 181-189, but if it is still not clear, let us know and we would be more than happy to continue the discussion.
>
> Reply #5: As summarized in the general response, UMFI values do not necessarily dominate MCI values. We removed this aspect in the revision. As for why UMFI does not share the flaw of giving non-zero importance to unrelated features, please see Reply #10, or refer to the proofs of UMFI satisfying the blood relation axiom in certain settings, found in Supplement C, as well as relevant experimental results (Subsection 4.1.4 and G.1.4).
>
> Reply #6: We agree with the reviewer’s comments on the marginal vs conditional framing to an extent. However, our intuition for viewing UMFI and MCI as marginal methods comes from papers by Gromping and Strobl, where marginal and conditional frameworks are said to define the two extremes for feature importance, and that they differ only in the face of dependent features. At one extreme, there is pairwise squared correlations (marginal) and at the other extreme, there is conditional permutation importance (conditional). Suppose that the feature set is composed of only two features, which are duplicates of each other, and they are correlated with the response with $r=0.5$. Then, the feature importance given by squared correlation is 0.25 for both features, but the importance given by conditional permutation importance is 0 for both features. If the evaluation function is $R^2$, both MCI and UMFI would give an importance of 0.25 to both features. Thus, MCI and UMFI act very similarly to the squared correlation, which is an extreme marginal method, and the way in which these methods treat correlated features is opposite to conditional permutation importance. This is a very interesting discussion and we would love to discuss this further, so feel free to argue with us on these points. We would like to make these distinctions more clear in our text.
>
> **Reviewer: Some potential issues with the experiments**
>
> Reply #7: We included the unmodified BRCA experiment in the supplement. Also, we included an experiment on a real dataset from hydrology in the supplement. Both of these are also in the revised version of the paper.
>
> Reply #8: The predictive performance of models is not a concern for us. The goal of the UMFI framework is to accurately rank features based on their association to the response within the data, not to optimize the model.
>
> Reply #9: In the supplementary material, we already examined many other baseline methods including ablation, permutation importance, and conditional permutation importance. In the revised version of the main text, we clarify that these baselines are tested in the Supplement (Lines 154-158).
>
> Reply #10: MCI giving high importance to randomized features is not a mistake. This is due to the fact that many ML models have poor estimation of dependence when given a small feature set, i.e. if $|S|=1$, then $\nu(S)<\nu(S,x)$ even if $x$ is a randomized feature. This issue was shown in the original MCI paper on page 9 of the MCI paper’s supplement. MCI gives highly significant importance to all features, whereas the baselines give zero or negative importance to about 5-15 of the features.

---

> ### Author Response · Authors · 2022-08-01
> **Response to Reviewer sCQt: Part 2**
>
> **Reviewer: The notation in this work was confusing at times**
>
> Reply #11: We believe we have resolved the notational issues in accordance with your suggestions in the resubmission.
>
> **Reviewer: Nits**
>
> Reply #12: Thank you for pointing out SPVIM to us, as we had not seen this paper before. We added this reference to the introduction/related works. From our understanding, SAGE is in fact a subset based method as can be seen in Covert et al. (2020) Section 3.3. You are correct that permutation importance is not a subset based method. We make this part of the introduction more clear in the updated text.
>
> Reply #13: It seems that Chen does not mention the word “marginal” in the “true to the data paper”. Were you thinking of another paper?
>
> **Reviewer: Questions**
>
> Reply #14: The domain and codomain of $g$ is not important, as long as $g$ can act on $F$. We have removed this from the resubmission to improve clarity.
>
> Reply #15: We think we have clarified all of the bullet points in this section in our above responses and the general response, but if anything is still unclear to you, please feel free to ask.
>
> **Reviewer: Limitations**
>
> Reply #16: We do not necessarily see any downsides to the example you laid out. As long as the variance contributions to the response are approximately equal, $x_1$ and $x_2$ should have around the same importance. But your intuition that the maximization framework could pose problems is correct. We address those in the general comments to all reviewers.
>
> References:
> Covert, I., Lundberg, S. M., & Lee, S. I. (2020). Understanding global feature contributions with additive importance measures. Advances in Neural Information Processing Systems, 33, 17212-17223.
>
> Debeer, D., & Strobl, C. (2020). Conditional permutation importance revisited. BMC bioinformatics, 21(1), 1-30.
>
> Grömping, U. (2009). Variable importance assessment in regression: linear regression versus random forest. The American Statistician, 63(4), 308-319.

---

> > ### Comment · Reviewer_sCQt · 2022-08-07
> > **Follow-ups**
> >
> > Thanks to the reviewers for their detailed response. I'm still examining the updates made on the theoretical side, but in the meantime there are a couple simpler topics to discuss. I'll use the authors' numbering of topics so it's clear what we're talking about.
> >
> > **Reply #6.** I see, I'm not as familiar with the terminology from Gromping and Strobl. In the two examples you mentioned, squared correlations and conditional permutation tests, I see that there's a significant difference between the two approaches. As a clarifying question, I'd ask where a (standard) permutation test falls on this spectrum - is it "marginal" or "conditional?" In the sense of how they deal with held-out features, I'm tempted to say a conditional permutation test is conditional whereas a standard permutation test is marginal, because one literally samples from the conditional distribution while the other samples from the marginal distribution. But I'm not sure what your categorization would say. When methods can vary in multiple dimensions (how they handle held-out features, and whether they consider a feature's effect in isolation or with all other features considered), I'm not sure the differences are adequately described by a single spectrum spanning marginal and conditional methods.
> >
> > I suppose the terminology I'm advocating for is best reflected in [1], where various feature importance methods are categorized across 3 axes: how they handle held-out features, what model behavior they focus on (loss vs. single prediction), and how they summarize each feature's contribution (e.g., deleting a single feature). Much of today's feature importance literature seems to be concerned with that first choice, where the key difference between methods is often whether held-out features are sampled from their conditional or marginal distribution (I can provide examples if helpful). This seems like a more concrete definition of conditional vs. marginal methods, but I now see that this isn't what the authors meant when alluding to the "conditional vs. marginal divide." I'm not sure what the resolution is, but after the authors response I still find that the framing of UMFI/MCI as marginal methods isn't very helpful and could at the very least be explained better.
> >
> > [1] Covert et al., "Explaining by removing: a unified framework for model explanation" (2021)
> >
> > **Reply #10.** I think I follow the authors' argument, but this prompts a follow-up question. For a random noise feature $x$, you could *maybe* observe $\nu(S, x) > \nu(S)$ for a very small dataset and when only examining in-sample (training) data, but the effect should disappear when evaluating the model on out-of-sample data. My follow-up is, were the experiments in the paper conducted using in-sample data? This seems problematic because training accuracy is heavily biased, and the large importance values for irrelevant features under MCI might vanish if the experiments were performed with out-of-sample data.
> >
> > **Reply #13.** Apologies, the authors are right about the paper not using the term marginal. There's been an unfortunate rebranding of sampling from the marginal distribution as an "interventional" approach, even though it's not causal and there are other methods that are actually are (see [2] based on the underlying causal graph in the data). [3] is a more recent paper that replaces the "interventional" term with "marginal." Anyway, what the authors call "interventional" in Chen et al. is in fact sampling held-out features from their marginal distribution, like how standard permutation tests do. So the trade-offs here are in fact between what I understand as "marginal" and "conditional" methods, i.e., those that sample held-out features from their marginal or conditional distributions.
> >
> > [2] Heskes et al., "Causal Shapley values: exploiting causal knowledge to explain individual predictions of complex models" (2020)
> >
> > [3] Chen et al., "Algorithms to estimate Shapley value feature attributions" (2022)
> >
> > **Reply #16.** The issue in the example I described is that $x_1$ is informative on its own and therefore (arguably) more important than $x_2$, which requires $x_3$ to be informative. For example, if we have a response variable $y$, then a noiseless feature $x_1 = y$ is more useful than a noisy feature $x_2 = y + \epsilon$ for $\epsilon \sim N(0, 100)$. However, $x_2$ yields a similar accuracy improvement to $x_1$ when it is introduced with $x_3 = y - \epsilon$ already present, so MCI/UMFI will assign $x_1$ and $x_2$ equal value. What do the authors think, do they agree that this seems undesirable?

---

> > > ### Author Response · Authors · 2022-08-08
> > > **Response to sCQt follow-ups**
> > >
> > > **Reply #6 and #13**
> > >
> > > Authors: Thank you for following up with us on this topic and referring us to some interesting papers. To answer your question about how we would classify standard permutation importance, please refer to Subsection G.1.3 in the Supplement, where we note that permutation importance is exactly in the middle of true-to-model and true-to-data methods with regards to its treatment of correlated features. This is because when two features are duplicates and equal to the response (let’s say $x_1=x_2=Y$), then, the squared correlation, MCI, and UMFI all give full importance to $x_1$ and $x_2$, permutation importance gives half of the importance to $x_1$ and half to $x_2$ (because $x_1$ will appear in approximately half the model and $x_2$ will be in the other half), and conditional permutation importance will give zero importance to both $x_1$ and $x_2$.
> > >
> > > We recognize the reviewer’s perspective of marginal vs. conditional in the distributional sense. **We have decided to relabel this division in the paper as true-to-data vs true-to-model** in order to completely disambiguate it from the marginal vs. conditional distributional divide as understood in other papers.
> > >
> > > **Reply #10**
> > >
> > > Authors: We agree that under most circumstances, out-of-sample data should be used to compute $\nu$. Indeed, in all of our experiments, we use out-of-sample data to compute $\nu$ (the out-of-bag error comes from out-of-sample data). Using out-of-sample data is a very important step if $\nu$ is assessed using machine learning models (e.g. random forests or extremely randomized trees) as the reviewer correctly points out that training accuracy can be heavily biased. However, even while using out-of-sample data, we may observe circumstances where $\nu(S,x)>\nu(S)$ when $x$ is random. If $S$ only contains one feature, there could be huge amounts of overfitting if the trained model is large (as is the case in random forests, xgboost, extremely randomized trees, etc…), leading to $\nu(S)$ being very small. Even if $x$ is completely random, it could reduce overfitting, leading to a bit of an increase in $\nu(S,x)$ compared to $\nu(S)$ when it is calculated using out-of-sample data. We have seen this happen in many experiments. We also note that out-of-sample data is not always needed. As reviewer qdSP pointed out, $\nu$ could be calculated with HSIC, which may not need out-of-sample data since no model is fit to the data, so overfitting is impossible. As a second example, suppose we could exactly calculate the mutual information $I(Y;S,x)$ and $I(Y;S)$, then again, no out-of-sample data would be needed.
> > >
> > > **Reply #16**
> > >
> > > Authors: We agree that it would be undesirable for $x_1$ and $x_2$ to be assigned equal value in the example given by the reviewer, but we disagree that MCI and UMFI would in fact assign $x_1$ and $x_2$ equal value. $x_1=y$, so $x_1$ would have maximal importance, given by $\nu(x_1)$, for both MCI and UMFI. Since $x_2=y+\epsilon$ and $x_3=y-\epsilon$, if we have both in a model, we would also predict $y$ perfectly, so $\nu(x_2, x_3)= \nu(x_1)$. As the variance of $\epsilon$ goes to infinity, we observe that $MCI(x_2) \to MCI(x_1)$, since $MCI(x_2)= \nu(x_2, x_3) - \nu(x_3) \to \nu(x_1) = MCI(x_1)$ since $\nu(x_3) \to 0$ as $var(\epsilon) \to \infty$.  So within MCI, the importance of $x_1$ is larger than the importance of $x_2$, but this difference can be made arbitrarily small. UMFI avoids this problem entirely because we preprocess the data by removing dependencies on the feature of interest. For instance, when removing dependencies on $x_2$ from $x_3$, the transformed variable $S^F_2$ would still have significant correlation with $y$, and hence, $U_\nu(x_2)=\nu(x_2,S^F_2)-\nu(S^F_2) < \nu(x_1)=U_\nu(x_1)$. After running simulated experiments following your example, we observed that UMFI gives large importance to $x_1$ and small importance to $x_2$ across different scales of variances for $\epsilon$. One way of seeing this is that since $\epsilon$ is random, it is not blood related to $y$, and hence has zero UMFI importance. Then, as the variance of $\epsilon$ increases, $x_2$ and $x_3$ become more and more similar to $\epsilon$, which reduces their respective UMFI scores.

---

> > > > ### Comment · Reviewer_sCQt · 2022-08-09
> > > > **Response (continued)**
> > > >
> > > > **Reply #6 and #13.** Apologies for making such a fuss about this, it's ultimately not a crucial part of the paper and just helps explain UMFI in the context of prior work. The resolution you've described sounds good to me, I suspect it will make the categorization of methods a bit easier to follow. (For example, I don't know how to make sense of permutation tests being exactly in the middle of "marginal" and "conditional" methods under the previous categorization.)
> > > >
> > > > One more nit I have about related work is that on lines 65-67, I don't see what makes SAGE or SPVIM less well-suited to explaining data relationships than MFI. Both papers talk explicitly about explaining real relationships in the data distribution. Perhaps SAGE suffers from its use of the marginal distribution in its approximation algorithm (although one could use a better conditional distribution estimate), but SPVIM does not have this issue (as it actually retrains the model with each feature subset).
> > > >
> > > > **Reply #10.** Thanks for explaining this, but this still doesn't sound right to me. A model trained on one feature does not seem capable of overfitting, this is much more likely to occur with high-dimensional data. (Note that this is why tree-based models use column subsampling as a means of regularization.) So I don't think I follow the argument for why a random noise feature $x$ should improve a model's out-of-sample performance, and I suspect that if you're observing any improvement in practice it could be due to your choice of hyperparameters for the original model. Furthermore, I think it's important to disentangle the real-world approximation procedure the theoretical version of the method based on mutual information: if we can calculate the true mutual information, there is surely no benefit to including a random noise feature in the model.
> > > >
> > > > **Reply #16.** Like you said, in the limit $var(\epsilon) \to \infty$, it seems like we would have $MCI(1) = max_S v(S \cup 1) - v(S) = v(1) - v(\emptyset) = v(2, 3) - v(3) = max_S v(S \cup 2) - v(S) = MCI(2)$. This means that at least MCI yields an undesirable outcome in this case. I guess I thought UMFI shared the flaw because I was thinking of the previous version of UMFI where we maximized over any preprocessing of the features. In this case, I believe the preprocessing that would have yielded the largest impact is $S_2 = x_3$ with $v(2, 3) - v(3)$. So it seems like the modified UMFI definition is a key factor here.
> > > >
> > > > **About the modified UMFI definition and theory.** The revisions made since the original reviews are substantial, and I haven't been able to revisit the revised paper with the same attention as the original version. The modifications to UMFI in section 3 do seem reasonable, but my conviction isn't 100%. (And based on an initial reading, it would be helpful to present definitions 1 and 2 in reverse order.) It seems like the main procedure has not changed, just the mathematical/theoretical characterization of the algorithm. Can the authors confirm whether this is correct?

---

> > > > > ### Author Response · Authors · 2022-08-09
> > > > > **Response to Reviewer sCQt**
> > > > >
> > > > > **Reply #6 and #13**
> > > > >
> > > > > SAGE, which seems fairly similar to SPVIM, is already compared to MCI in the original MCI paper and they show that MCI outperforms SAGE in several experiments. SAGE and SPVIM do not follow the elimination axiom, so adding a feature to the feature set could decrease the importance of other features, which does not make sense in many contexts (Figure 1 of the MCI paper).
> > > > >
> > > > > **Reply #10**
> > > > >
> > > > > If you have access to R you can run the following code:
> > > > >
> > > > > ```
> > > > > library(ranger)
> > > > > nobs=1000
> > > > > #with extra noise feature
> > > > > dat<-data.frame(x=rnorm(nobs),x2=rnorm(nobs))
> > > > > dat$y=dat$x+rnorm(nobs)
> > > > > mod<-ranger(y~.,data=dat)
> > > > > mod$r.squared #about 0.44
> > > > >
> > > > > #with only one feature
> > > > > mod<-ranger(y~x,data=dat)
> > > > > mod$r.squared # about 0.35
> > > > > ```
> > > > >
> > > > >
> > > > > We are not doing any special hyperparameter tricks here. We are just using default hyperparameters as picking hyperparameters for each of the 1000s of models is fairly impractical. We showed in our Supplemental experiments that this can also happen with extremely randomized trees. We agree that we need to do a better job of disentangling the downsides of real-world estimates of MCI vs theoretical idealized estimates of MCI. In both cases (real-world and theoretically) MCI can give importance to features that are completely unrelated to the response. We can see this in the real-world from the example code provided above and in several of our experiments (Subsection 4.1.3 and 4.2). We see this theoretically in the downsides of the marginal contribution axiom when faced with data generated from the causal graph $Y \gets S \to G \gets E$ as pointed out by Harel et al., 2022 and discussed in Supplement B.
> > > > >
> > > > > **Reply #16**
> > > > >
> > > > > You are correct.
> > > > >
> > > > > **About the modified UMFI definition and theory.**
> > > > >
> > > > > We are happy to reverse the order of definitions 1 and 2 in the revised text, which will be posted shortly. You are correct in saying that none of the procedures for UMFI have changed. We have only altered the mathematical characterization of UMFI by removing the maximization connection with MCI, and have updated the theoretical justifications surrounding UMFI. Because none of the procedures for UMFI have changed, none of the experimental results change, though we do add an additional experiment to emphasize the use of the blood relation axiom (Subsection 4.1.4).

---

> > > > > > ### Comment · Reviewer_sCQt · 2022-08-09
> > > > > > **Response**
> > > > > >
> > > > > > **Reply #6 and #13.** I'm not claiming that SAGE/SPVIM will necessarily achieve better performance than MCI/UMFI, but it's inaccurate and unfair to the authors to claim that there are no other methods designed to explain importance in the true data distribution. With SPVIM for example, that's precisely what it was designed to do.
> > > > > >
> > > > > > **Reply #10.** Thanks for providing a speedy example, but the code you've shared is based on in-sample $R^2$ so I'm not sure it addresses what we've been talking about. On the other hand, I see what you're saying about the collider scenario - that seems like a valid example of an unimportant feature that receives non-zero importance under MCI. It's somewhat up for debate whether $E$ is truly uninformative though, because it's not independent from $Y$ under arbitrary conditioning, but I can see why you would want to assign it zero importance. Still, your argument for why an independent random noise feature should improve out-of-sample performance doesn't sound right, and the example provided doesn't support it. $I(Y; S, x) = I(Y; S)$ if $x$ is independent random noise, so any difference you observe in practice may be an artifact of the training procedure.
> > > > > >
> > > > > > **Modified UMFI.** Sounds good, I'm going to raise my score then. I still think the issue above (Reply #10) is a concerning aspect of the experiments. It's slightly mitigated by the fact that MCI observed the same issue in the original paper, but I would ask the authors to think about this more carefully to avoid writing something misleading in the paper.

---

> > > > > > > ### Author Response · Authors · 2022-08-09
> > > > > > > **Response to Reviewer sCQt**
> > > > > > >
> > > > > > > **Reply #6 and #13.**
> > > > > > >
> > > > > > > Ah, we see what you are saying now. Thank you for catching this. We will ensure that this wording is changed for the camera-ready version if we get accepted.
> > > > > > >
> > > > > > > **Reply #10**
> > > > > > >
> > > > > > > The code that we shared does not show in-sample $R^2$. See page 22 of the documentation of the ranger package. The “r.squared” output is based on out-of-bag data. For a detailed explanation on what out-of-bag $R^2$ is measuring, see Section 3.1 of Breiman, 2001. As Breiman notes, the out-of-bag error is just as accurate as a set aside test set. Indeed he states “Therefore, using the out-of-bag error estimate removes the need for a set aside test set.”. We note that if one looks at the procedure for out-of-bag error estimation, the out-of-bag $R^2$ (which we use throughout our paper) is similar to cross validation. Therefore, it is certainly an error estimate on out-of-sample data. Further we can look to the following code example to show that the “r.squared” parameter is not the same as the in-sample $R^2$ (it is a bit stochastic, so run it a few times):
> > > > > > >
> > > > > > > ```
> > > > > > > library(ranger)
> > > > > > > nobs=1000
> > > > > > > #with extra noise feature
> > > > > > > dat<-data.frame(x=rnorm(nobs),x2=rnorm(nobs))
> > > > > > > dat$y=dat$x+rnorm(nobs)
> > > > > > > mod<-ranger(y~.,data=dat)
> > > > > > > mod$r.squared # about 0.44
> > > > > > > insamp_preds<-predict(mod,dat)$predictions
> > > > > > > insamp_R2<-1-sum((insamp_preds-dat$y)^2)/sum((dat$y-mean(dat$y))^2)
> > > > > > > insamp_R2 # about 0.87
> > > > > > >
> > > > > > > #with only one feature
> > > > > > > mod<-ranger(y~x,data=dat)
> > > > > > > mod$r.squared #about 0.35
> > > > > > > insamp_preds<-predict(mod,dat)$predictions
> > > > > > > insamp_R2<-1-sum((insamp_preds-dat$y)^2)/sum((dat$y-mean(dat$y))^2)
> > > > > > > insamp_R2 # about 0.86
> > > > > > > ```
> > > > > > > We must clarify one point. The collider example and the property where random noise can increase out-of-sample accuracy are separate issues, but they lead to the same overarching issue (MCI overestimates the importance of features unrelated to the response). The out-of-sample accuracy issue only happens in practice for many machine learning algorithms when the size of the feature set is very small (see code above), but as you pointed out, if we look to theory (i.e., use mutual information to calculate $\nu$), the problem disappears. You correctly attributed this discrepancy to the training procedure, but this is still a major issue, as we have seen this issue across all popular statistical learning algorithms (random forests, extremely randomized trees, and xgboost), even when we use arbitrary hyperparameters and out-of-sample data to estimate predictive power. On the other hand, the collider issue exists in both theory (see Supplement B and Harel et al., 2022) and in practice (see our paper, Section 4.1.4).
> > > > > > >
> > > > > > > **Modified UMFI.**
> > > > > > >
> > > > > > > Thank you for raising your score, and more importantly, we thank you for having such an in-depth conversation with us. Your suggestions have certainly made our paper better.
> > > > > > >
> > > > > > > **References**
> > > > > > >
> > > > > > > https://cran.r-project.org/web/packages/ranger/ranger.pdf
> > > > > > >
> > > > > > > Breiman, L. (2001). Random forests. Machine learning, 45(1), 5-32.

---

> > > > > > > > ### Comment · Reviewer_sCQt · 2022-08-09
> > > > > > > > **Thanks**
> > > > > > > >
> > > > > > > > Sounds good, I think this resolves all my concerns. And yes I think discussing these points and making the changes from the previous posts will improve the paper. Best of luck.

---

### Official Review · Reviewer_rqHi · 2022-07-11

**Rating:** 6
**Confidence:** 3
**Soundness:** 3 good
**Presentation:** 2 fair
**Contribution:** 2 fair

**Summary:**

This paper proposes an ultra-marginal feature importance (UMFI) method by extending marginal conditional importance (MCI) methods for evaluating feature importance. Some experiments have been conducted on simulated and real-world data sets.


**Questions:**

See above.

**Limitations:**

Limitation statements are given.


**Strengths And Weaknesses:**

This paper studies an important problem. The authors have reviewed the related work well.

The extension of the work from MCI methods is by introducing an information subset in feature importance evaluation. The authors do not justify why an information subset is necessary. I have a few questions about the proposed method.

When evaluating the feature impotence of f, there needs to find the largest information subset S*, such that f and S* are independent. My question is if there is no such an S*, how the importance of f is evaluated? This is real when all features are correlated.

I need to be convinced to use the largest information subset S* in evaluating the importance of f. I use a causal effect estimation to understand the solution since feature importance is related to the causal effect of f on Y. When S* is independent of f, it does not ``confound" f in a viewpoint of causal effect estimation. Why should we remove the effect of S* on Y when estimating the effect of f on Y?

By reading the algorithm, the importance of a feature is related to a model, see Lines 3-5. Do authors mean to explain feature importance in the model? This is not clear in the paper.

The predictions of a model depend on the parameters in practice. Based on Lines 4 and 5 in Algorithm 1, the importance of a feature can be dependent on the parameters. How do authors avoid the effect of parameter variation on evaluating the importance of a feature?

---

After discussions.

The authors answered my queries, and their revisions have addressed my major concerns, I raise my overall rate.

---

> ### Author Response · Authors · 2022-08-01
> **Response to Reviewer rqHi**
>
> Reply #1: We thank the reviewer for their questions and suggestions. Overall, it seems that the main issues pointed out by this reviewer come from the theoretical framework presented in Section 3 of the original submission . To ensure that the theory is equally as strong as the experiments or even stronger, we have rewritten Sections 2 and 3 to make our framework more rigorous and substantive.
>
> **Reviewer: The authors do not justify why an information subset is necessary.**
>
> Reply #2: Information subsets are introduced in order to allow for independent representations of data, which are transformations of the feature set, and therefore not generally part of the raw feature set. These concepts are tied together explicitly in Section 3 of the revision. The integration of independent representations of data into our method enables UMFI to have some desirable properties, as proved in Supplement C, as well as better performance, as shown in the experiments.
>
> **Reviewer: If there is no such an $S^\*$, how is the importance of f evaluated? This is real when all features are correlated.**
>
> Reply #3:  In the setting where all features are perfectly correlated, we note that $S^\*$ would be a constant predictor bearing no information about $Y$, and hence the importance of $x$ would be given by $\nu(x)$. It is difficult to say if the optimal preprocessing will always exist, however, we prove in the Gaussian case that it exists via linear regression (see Theorem C.1 in the Supplement). Further, UMFI is always approximated in practice, so even if an optimal preprocessing does not exist, we approximate something close to it. In our experimental results, we have shown that UMFI produces very good results even though the preprocessings were not optimal.
>
> **Reviewer: I need to be convinced to use the largest information subset $S^\*$...**
>
> Reply #4: We do not remove the effect of $S^\*$ on $Y$ when estimating the effect of feature $f$ on $Y$. As shown in Definition 2, we effectively measure the difference in prediction power towards $Y$ of $f$ on top of $S^\*$.  We strongly agree that feature importance should be related to the underlying causal graph as well. This motivated our introduction of the blood relation axiom in Section 2, which stipulates that if data is generated from a causal graph, then a feature importance method should give non-zero importance to a feature $f$ if and only if $f$ is blood related to $Y$ in the causal graph. We note that a feature $x$ being blood related to $Y$ is equivalent to $x$ being statistically associated with $Y$ (Williams et al. 2018). The formulation of UMFI with $S^\*$ enables it to satisfy the blood relation axiom in various settings (see Supplement C and Subsection 4.1.4 in the revision). The other tested metrics, which include MCI, ablation, permutation importance, and conditional permutation importance, all fail in this respect (see Supplement G.1.4 and Section 4.1.4).
>
> **Reviewer: By reading the algorithm, the importance of a feature is related to a model…**
>
> Reply #5: No, the goal of UMFI is to explain hidden relationships in data. We have made several revisions to make this more clear. First, as pointed out by Reviewer qdSP, a model need not even be involved in UMFI. All that is required is a measure of dependence (e.g., HSIC or mutual information). We have clarified this point in the revised version of Section 3. Second, even when UMFI is implemented with models, we train two independent models, so if one were to argue that our feature importance method explains a model, we would have to ask “which model?”. Instead, UMFI explains the data because we can learn something about the causal graph from it. We have clarified this with our axioms in Section 2, further discussion in Supplement A.2, and corresponding proofs in Supplement C.
>
> **Reviewer: The predictions of a model depend on the parameters in practice…**
>
> Reply #6: Indeed, predictions and model accuracy can change due to slight changes in model parameters due to different data samples or just due to different random seeds. In our experiments, we fight against this possible limitation by finding the median importance value over many estimates of the feature importance, which is practical due to the moderate time complexity required for computing UMFI scores. In the simulation studies in subsection 4.1, we calculate the feature importance 100 times, and in the real data study, presented in subsection 4.2 and Supplement G.2, we calculate the feature importance 200-5000 times. Indeed, we would not recommend someone to put great trust in a single estimate of UMFI. Instead, calculating UMFI and estimating its median over many initializations would be wise. We emphasize this point more in our revised text (Lines 143-144).
>
> References:
>
> Williams, T. C., Bach, C. C., Matthiesen, N. B., Henriksen, T. B., & Gagliardi, L. (2018). Directed acyclic graphs: a tool for causal studies in paediatrics. Pediatric research, 84(4), 487-493.

---

> > ### Comment · Reviewer_rqHi · 2022-08-08
> > **Re: Response to Reviewer rqHi**
> >
> > Thanks for answering my queries and revising the paper.
> >
> > I have some follow-up questions (discussions).
> >
> > “ Information subsets are introduced in order to allow for independent representations of data,  which are transformations of the feature set, and therefore not generally part of the raw feature set. ”
> >
> > Transformation is very important in UMFI. My question is whether it is done for each $X_i$? If so, will the different transformations for different features lead to inconsistency in feature importance estimation of different features?
> >
> > Transformation should be explained briefly in the main text and also be included in the algorithm. So readers can understand the complete process of UMFI.
> >
> >
> >
> > “ This motivated our introduction of the blood relation axiom in Section 2, which stipulates that if data is generated from a causal graph, then a feature importance method should give non-zero importance to a feature f if and only if f is blood related to Y in the causal graph.”
> >
> > The authors’ Blood relation axion is likely incorrect. “Two vertices in a causal graph are said to be blood related if … or if there is a backdoor path between them via a common ancestor.” For example, in the causal graph $X_1 \to X_2 $ and $X_1 \to Y$, $X_2$ is blood related to Y. Note that there is not a causal relationship between $X_2$ and $Y$ since there is no edge between them. We usually say that the relationship between them is spurious. I do not believe that such a relationship should be called a blood relationship. I understand that the authors may argue that $X_2$ is necessary when $X_1$ is unmeasured. However, this is not clear in the axion.
> >
> > Another note: I do not agree with the authors' explanation of the collider example of Harel et al. [25]. In the causal graph, $Y \leftarrow S \to G \leftarrow E$ where $S$ is unmeasured (Lines 83-90 in the Appendix). The Authors think that the feature importance of $G$ on $Y$ should be zero since $G$  has no causal relationship with $Y$. Here, the role of $E$ is the same as the role of $X_2$ in the above example since they both do not have causal relationships with $Y$, but are proxies for un-measured causes of $Y$, i.e. $X_1$ and $S$. So, the feature importance of $G$ on $Y$ should not be zero. If $G$'s feature importance is zero, which variable explains $Y$ in the graph?
> >
> > I suggest that authors remove causal discussions since causal definitions need quite a few assumptions and basic concepts, for example, $d$-separation and backdoor paths. Without the assumptions and basic concepts, discussions can be confusing.
> >
> >
> >
> >  “in the real data study, presented in subsection 4.2 and Supplement G.2, we calculate the feature importance 200-5000 times. Indeed, we would not recommend someone to put great trust in a single estimate of UMFI. ”
> >
> > It will be helpful to discuss the sources of the variability. My question is what randomisation mechanism ensures the median feature importance to approximate the optimal estimation?
> >
> > The authors do not recommend the single estimate of UMFI because of its higher variability than MCI, and hence the multiple iterations should be included in the algorithm to make this clear.
> >
> > In figure 3 of computational time, what is the number of iterations being considered in the time calculation of UMFI? Has the time for transformations been included?

---

> > > ### Author Response · Authors · 2022-08-08
> > > **Response to rqHi follow-ups**
> > >
> > > **Transformations**
> > >
> > > When evaluating the importance of feature $x_i$, we seek to transform the feature set to be independent of $x_i$ while minimizing distortion. Different features of interest $x$ will necessitate different transformations $S^F_x$. We have no reason to believe that importance scores for different features will lead to inconsistent importance scores. In fact, we have demonstrated theoretically and experimentally that UMFI consistently exhibits desirable properties. Transformations are discussed in Lines 118-130, and implementation details for optimal transport and linear regression are presented in Supplement E. Line 2 of Algorithm 1 details the need for a transformation, but we do not assert a specific method in this line of the algorithm because UMFI is just a framework. There are many different algorithms one can use to transform the feature set, so we leave this part open to the user.
> > >
> > > **Blood relation axiom**
> > >
> > > Given the causal graph $Y \gets X_1 \to X_2$, $Y$ is blood related to $X_2$ because, when viewed as a family tree, $Y$ and $X_2$ are siblings as they share a parent $X_1$. We argue that $X_2$ must be given non-zero importance for several reasons. First, as you correctly point out, $X_1$ may not be measured, so in this case, $X_2$ should be assigned importance. Second, in genome-wide association studies, scientists want a list of all genes associated with some disease. Indeed, two variables are statistically associated iff they are blood related (see Williams et al., 2018, Greenland et al. 1999). Third, in Earth sciences, we often rely on proxy variables that do not directly cause the response, but that are blood related to the response. For example, in hydrology, we know that low flow characteristics are directly caused by snow fraction. However, snow fraction is not reliably globally available, so snow persistence (which is caused by snow fraction) can be used as a valid proxy because it can be easily measured from satellite observations. If one were to use a dataset with both features, it would be misleading to conclude that snow persistence is not important because it may lead researchers to not consider snow persistence as a valid proxy in future work. Finally, even if one believes that $X_2$ should have zero importance in the graph $Y \gets X_1 \to X_2$, this graph is Markov equivalent to the causal graph $Y \gets X_1 \gets X_2$, so when we have access to the data, but not the casual graph, $X_2$ must be found to be important in an explanatory setting (see Section 7 of Gromping, 2009).
> > >
> > > **Collider**
> > >
> > > We did not say that the importance of $G$ should be zero. We said that $E$ should have zero importance given the causal graph $Y \gets S \to G \gets E$. Indeed, $G$ is blood related to $Y$ through their common parent $S$. So by the blood relation axiom, $G$ should be given positive importance, but $E$ is not blood related to $Y$ and should be given 0 importance. $G$ inherently contains information about $Y$, but this information is noised up by $E$. Therefore, although $E$ can be used to denoise $G$ and predict $Y$ better, only $G$ should be given importance when explaining the data, and indeed, only $G$ is blood related to $Y$. We have revised this passage of the Supplement to clarify our reasoning.
> > >
> > > We believe that the causal discussions add significant value to the paper and that UMFI obeying the blood relation axiom under many circumstances both theoretically and experimentally is a beautiful and impactful result. We include all relevant assumptions for discussing causality in Supplement C, and in the revised text, we refer the reader to additional papers for more information about relevant causal graph concepts.
> > >
> > > **Variability**
> > >
> > > The variability shown in the experiments comes from the stochasticity of ML methods. Additional variability appears due to subsampling the data at each iteration. We use the median because UMFI can give exactly zero importance, and taking the median of 200 iterations where 190 of them are 0 and 10 of them are 0.001 produces clear results.
> > >
> > > We agree that the usage of multiple iterations should be emphasized in Section 3, and we have revised the text accordingly. We did not revise this in the algorithm since UMFI can also be computed using methods such as HSIC (see reviewer qdSP), which could result in little-to-no variability.
> > >
> > > In Figure 3, the importance of each feature in the current dataset is calculated once and have revised the caption. The time for transformations is included in this experiment, as stated in lines 253-261.
> > >
> > > **References**
> > >
> > > Greenland, S., Pearl, J., & Robins, J. M. (1999). Causal diagrams for epidemiologic research. Epidemiology, 37-48.
> > >
> > > Grömping, U. (2009). Variable importance assessment in regression: linear regression versus random forest. The American Statistician, 63(4), 308-319.
> > >
> > > Williams, T. C., et al. (2018). Directed acyclic graphs: a tool for causal studies in paediatrics. Pediatric research, 84(4), 487-493.

---

### Official Review · Reviewer_3eYw · 2022-07-11

**Rating:** 7
**Confidence:** 3
**Soundness:** 3 good
**Presentation:** 3 good
**Contribution:** 3 good

**Summary:**

The authors provide a simple yet effective extension of the recently proposed marginal feature importance criterion that allows a fast and accurate estimate of the marginal importance of each feature in the presence of feature correlation and non-linear interactions.

**Questions:**

In the proof of Theorem 3.1, you seem to assume that the maximizer of eq. 3 is unique. But what if \tilde{S}^* is also a maximizer of 3? I guess for your scope it is enough that S^* is one of the maximizers, but this should be clarified for the sake of soundness.



**Limitations:**

The limitations of the approach are exhaustively discussed in the conclusion on the paper.

**Strengths And Weaknesses:**

Strengths

A simple but clever extension of the marginal feature importance criterion allows to circumvent the need for enumerating all potential feature subsets, substantially increasing the practical applicability of the approach.

Weaknesses:

The main ingredients were already all in the original paper introducing marginal feature importance. From this perspective the novelty of the work is not dramatic.

There is a lack of intuition in introducing the ultra-marginal feature importance criterion. It would be useful to discuss the role of g and its use in dependency removal when introducing it, possibly providing an illustrative example. Overall, the methodological section is very shallow and should be expanded to better highlight the rationale and relevance of the contribution.

Minor:

In theorem 3.1, S should be orthogonal to f, not to F.

I suggest to better contextualize the related work or move it to the end of the paper, as it's unclear why certain topics are being discussed (e.g. orthogonal predictors).

AFTER REBUTTAL
The authors improved the clarity of manuscript and better highlighted the significance of the contribution.

---

> ### Author Response · Authors · 2022-08-01
> **Response to Reviewer 3eYw**
>
> Reply #1: We thank the reviewer for pointing out some weaknesses in the paper. We agree with many of the points that the reviewer made, and provide a revised version of the text that improves upon these weaknesses.
>
> **Reviewer: The main ingredients were already all in the original paper introducing marginal feature importance. From this perspective the novelty of the work is not dramatic.**
>
> Reply #2: Due to finding more examples where MCI does not accurately describe data, but UMFI does, we move further away from the original MCI method in the revised version of the text. In particular, we explain that MCI’s marginal contribution axiom is at odds with providing appropriate importance scores in some causal settings (See general response to all reviewers), and this is explored experimentally in subsection 4.1.4. With these clarifications, the addition of our newly proposed axioms, and corresponding proofs provided in Supplement C, we believe that our paper presents novel insights about explaining relationships in data using feature importance.
>
> **Reviewer: There is a lack of intuition in introducing the ultra-marginal feature importance criterion. It would be useful to discuss the role of g and its use in dependency removal when introducing it, possibly providing an illustrative example. Overall, the methodological section is very shallow and should be expanded to better highlight the rationale and relevance of the contribution.**
>
> Reply #3: We agree that the methodological section was lacking and have significantly altered it in the new revision. This includes an explicit definition of optimal preprocessings (achieved by a function $g(F)$). Illustrative examples of such functions $g$ used for dependency removal can be found in Supplement E. Further, we improve the theoretical motivations behind UMFI with axioms in Section 2 and proofs of some desired properties in Supplement C.
>
> **Reviewer: I suggest to better contextualize the related work or move it to the end of the paper, as it's unclear why certain topics are being discussed (e.g. orthogonal predictors).**
>
> Reply #4: We agree that the previous formulation of the related work was slightly confusing. In the updated paper, we combine the MCI and related works section and make it more clear how orthogonal predictors relate to our work. In essence, we aim to find a representation of the data that is independent of some other feature. Orthogonality is a weaker version of independence, thus the two are related. We add this explanation to make the link more clear (Lines 69-74).
>
> **Reviewer: In the proof of Theorem 3.1, you seem to assume that the maximizer of eq. 3 is unique. But what if $\tilde{S}^\*$ is also a maximizer of 3? I guess for your scope it is enough that $S^\*$ is one of the maximizers, but this should be clarified for the sake of soundness.**
>
> Reply #5: We agree that it is important to clearly explain the non-uniqueness of $S^\*$, and we believe we did a poor job of explaining this. In our revised version of the text, we provide a rigorous definition for an optimal preprocessing $S^\*$ (now $S^F_x$) in Definition 1, and we specify its non-uniqueness in multiple parts of the paper (Lines 120-132).

---

### Author Response · Authors · 2022-07-30
**General response to all reviewers: Changes to theoretical framework of UMFI**

We thank the reviewers for their thoughtful comments. In this reply, we will address the theoretical issues laid out by several of the reviewers. Please see the revised version of the text that we just submitted.

Many of the reviewers pointed out that our initial submission had weak theoretical justifications. For example, 3eYw states that “there is a lack of intuition in introducing” UMFI, and the methods section is “very shallow and should be expanded to better highlight the rationale and relevance”. Also, rqHi is unsure if we are truly explaining the data and states that we “do not justify why an information subset is necessary”. **We have deepened the theory of UMFI by introducing three axioms, which we argue capture intuitive and useful properties for explaining the data. These axioms are presented in Section 2 of the resubmission, and proofs of UMFI satisfying these axioms are provided in Supplement C.** We note that dependency removal is critical to the proofs of UMFI satisfying the desired axioms.

Harel et. al (2022), coauthored by one of the authors of the MCI paper, demonstrate inherent problems with the marginal contribution axiom, which was one of the axioms introduced in the MCI paper. The key example to consider is the causal graph with a collider (C<-S->G<-E) given in Subsection 3.3 in Harel et. al (2022). Because of this issue, we choose to improve the MCI framework, rather than generalize it, by accepting the elimination axiom and a generalization of the duplication invariance property from the MCI paper as our first two axioms and by replacing the marginal contribution axiom with our blood relation axiom. **With the blood relation axiom, we prove that UMFI can detect part of the structure of the underlying causal graph, and UMFI gives non-zero importance iff the feature is statistically associated with the response.**


Three out of the four reviewers questioned our use of S*. 3eYw questions its uniqueness, rqHi questions its existence, and sCQt questions the definition and rigor of removing dependencies and S*. In the new revision, we provide an explicit definition for an optimal preprocessing S* (see Definition 1 of revised text). We proved the existence of optimal preprocessings in the multivariate normal setting (see Supplement C in the revised text). Although we are interested in proving the existence of optimal preprocessings in broader settings, non-existence is not typically a concern in practice, since the experimental results demonstrate that UMFI achieves strong results when computed using non-optimal preprocessings. We also clarify that optimal preprocessings, when they exist, are not unique. Indeed, we may adjust optimal preprocessings via constant factors without violating independence or the information content. This is further clarified in Section 3 of the revised text.


sCQt correctly points out that we did not properly justify how UMFI can better detect unrelated features if UMFI is “strictly larger” than MCI by solving a more general maximization problem. **This is an important criticism, as it turns out that the actual computed UMFI score for a feature $x$, given by $U^{F,Y}_\nu(x)=\nu(S^F_x,x)-\nu(S^F_x)$ ($S^F_x$ was previously denoted $S^\*$) does not necessarily solve Equation (3) of our paper, and therefore UMFI does not generalize the maximization problem from MCI.** Although no reviewers commented on this, we realized that **our proof of Theorem 3.1 was flawed since we misused monotonicity in line 138 of the original submission**. Indeed, removing information pertaining to a feature $x$ from a set of features $A$ may increase the mutual information between $A$ and the response $Y$. **We apologize for this mistake, and we remove the maximization formulation of UMFI in our new revision, clarifying that UMFI is defined by the direct computation $U^{F,Y}_\nu(x)=\nu(S^F_x,x)-\nu(S^F_x)$.**

Although we found that UMFI does not generalize the framework of MCI as closely as we first expected, **we emphasize that this does not alter the previous experimental results in any way, since we always use the direct computation, $\nu(S^F_x,x)-\nu(S^F_x)$, rather than the solution to the maximization problem in Equation (3). In fact, the theoretical and experimental results are improved with new axioms and an additional causality experiment that shows that UMFI is superior to MCI and other baseline methods.** The true formulation for UMFI is given by $U^{F,Y}_\nu(x)=\nu(S^F_x,x)-\nu(S^F_x)$ and Equation (3) in the original text was only meant to link UMFI with MCI, so removing this link does not hinder our paper, in fact, moving further away from MCI was seen as a potential improvement by 3eYw. **UMFI remains a strong true-to-the-data importance metric that performs better than competing methods across diverse settings, while requiring a fraction of the runtime of MCI.  In all, we believe that the resubmission greatly improves upon the theoretical justifications for UMFI.**

---

### Author Response · Authors · 2022-07-30
**General response to all reviewers: Additional Experiments**

We thank the reviewers for their thoughtful comments. In this reply, we will address the experimental issues laid out by several of the reviewers. Please see the revised version of the text that we just submitted.

Several reviewers pointed out the need for additional experiments. For example, sCQt requests us to “present additional results with the unmodified BRCA data”.  This was already done in the original submission (Supplement G.3). Others also suggested additional experiments with different evaluation functions $\nu$ and additional baselines. In a question relating to our experiments, rqHi asks us how we “avoid the effect of parameter variation on evaluating the importance of a feature” when UMFI is used in practice, sCQt requested that we should “include other baselines'', and qdSP asks "it's quite natural to wonder how critical the choice of supervised learning methods is: have this been explored?". This was also already done in the supplement under Appendix G by rerunning the experiments with extremely randomized trees instead of random forests for comparing MCI vs UMFI and running the same experiments over permutation importance, conditional permutation importance, and ablation for further baseline comparisons.

To clarify why MCI fails to detect correlated interactions as requested by sCQt, we slightly change the nonlinear interaction study to make it more comparable to the correlated interaction study. Further, though it was not directly requested, we decided to run an additional simulation study based on the collider example from Harel et. al (2022) to test the ability of methods to return importance scores that are consistent with the causal structure of the data. This was first tested on MCI, UMFI_LR, and UMFI_OT in Section 4.1, and then on additional methods in Supplement G. **The results show that both implementations of UMFI succeed in giving non-blood related features 0 importance while giving significant importance to blood related features.** We note that this experiment was performed in a synergistic and non-Gaussian setting, which further demonstrates the power of UMFI to obey the blood relation axiom outside the scope of the conditions that we have proved thus far. The other tested metrics, which include MCI, ablation, permutation importance, and conditional permutation importance, all fail in this respect. **In all, we believe that the resubmission makes the experimental evidence supporting UMFI even stronger.**


Harel, N., Gilad-Bachrach, R., & Obolski, U. (2022). Inherent Inconsistencies of Feature Importance. arXiv preprint arXiv:2206.08204.

---

### Comment · Area_Chair_dCAh · 2022-08-08
**MCI limitations**

This is question to the authors.

 In the introduction (lines 50-54) three limitations of MCI are presented. While the first one is provided with a supporting argument (the time complexity is due to the need to retrain the model), the other limitations are provided without such a supporting argument. Can you please add the justification for these statements? This can be a reference to another paper that already demonstrated it, or otherwise an example or a mathematical proof.

---

> ### Author Response · Authors · 2022-08-09
> **Reply to Area Chair dCAh**
>
> Thank you for the suggestion. We will add justifications for the other limitations in the introduction and upload the revised paper shortly. To be abundantly clear, we will explain the shortcomings of MCI in this comment as well.
>
> **Limitation of MCI #2: Second, although it can handle complex feature interactions and data with correlated features, MCI underestimates the importance of correlated features that form interaction effects.**
>
> We are the first to point out this shortcoming of MCI, so we will explain this issue with a simple example. MCI underestimates correlated features that form interaction effects because if a feature $x_2$ is highly correlated with the feature of interest $x_1$, then it is unlikely that the correlated feature will appear in the maximizing subset for MCI when calculating the importance of $x_1$ even if $Y$ contains an interaction effect with both $x_1$ and $x_2$.  This is because the additional predictive power offered by $x_1$ on top of a subset $S$ would be diminished by the presence of $x_2 \in S$. Evidence for this limitation is further supported by the experiment in Subsection 4.1.2.
>
> **Limitation of MCI #3: Third, MCI can give non-zero importance to features that are completely unrelated to the response variable.**
>
>
> First, we note that evidence for MCI giving non-zero importance to features that are completely unrelated to the response is provided on page 9 of the Supplement of the original MCI paper (Catav et al., 2021). In Figure 3, MCI is shown to give non-zero importance to all 50 genes even though 40 of the genes are randomly selected from the genome, so it is extremely unlikely that all 50 of these genes should be important. In practice, this happens because many ML models poorly estimate dependence when given a small feature set due to overfitting. This can result in $\nu(S)<\nu(S,x)$ when $|S|=1$ even if $x$ is a randomized feature. This issue is further justified in theory by issues arising from the marginal contribution axiom, as discussed in Supplement B. Indeed, MCI may give importance to completely unrelated features, as pointed out by the collider example $Y \gets S \to G \gets E$ in Harel et al. (2022). $E$ has no relationship with $Y$, however, if $F=${$G,E$}, then the ablation score $A_\nu(E)=\nu(G,E)-\nu(G)>0$, since $E$ can be used to denoise $G$ and gain information about $S$ and $Y$. Then, by the marginal contribution axiom, which is one of the three axioms that MCI is based on, $E$ is given non-zero importance by MCI. Evidence for this limitation is further supported by the experiments in Subsections 4.1.3, 4.1.4, and 4.2.
>
> **References**
>
> Harel, N., Gilad-Bachrach, R., & Obolski, U. (2022). Inherent Inconsistencies of Feature Importance. arXiv preprint arXiv:2206.08204.
>
> Catav, A., Fu, B., Zoabi, Y., Meilik, A. L. W., Shomron, N., Ernst, J., ... & Gilad-Bachrach, R. (2021, July). Marginal contribution feature importance-an axiomatic approach for explaining data. In International Conference on Machine Learning (pp. 1324-1335). PMLR.

---

### Meta-Review · Area_Chair_dCAh · 2022-08-21

**Recommendation:** Reject
**Confidence:** Less certain

**Metareview:**

This work makes a significant contribution to establishing the theoretical foundations for feature importance. The authors suggest a set of axioms that a feature importance score should have and introduce an algorithm that computes a feature importance score that has these required properties. In addition to the theoretical work, a compelling empirical evaluation is conducted showing significant improvement over previous results.

After a good discussion between the reviewers and the authors and improvements to the paper introduced due to this discussion, the result is a good paper that is of great interest to the NeurIPS community. However, the added content also raised some concern about the accuracy of some statements, especially with respect to the blood relation. The main concern is that it is not clear that the algorithm provided has the blood relation property. Moreover, it is not clear that it is possible to fulfil this relation. Here are two scenarios that may be problematic:
1. In the fully observed setting if X is a confounder of Y and Z while Z is identical to X then X blocks the backdoor from Y to Z and therefore, according to the blood relation axiom the importance of Z should be zero while the importance of X should be positive since it has direct causal relation with Y. However,  the roles of X and Z are indistinguishable and therefore it might as well be that Z is the confounder and therefore should have a non-zero importance.
2. In the partially observed setting, if S is an unobserved uniformly distributed integer in the range 1..8, Y is the sign of S, and X is an indicator of S being greater then 4 the according to the blood relation axiom, since there is a backdoor between Y and X when S in unobserved then the importance of X should be non-zero. However, this setting is indistinguishable from the setting in which X and Y are uncorrelated Bernoulli variables in which case the importance of X should be zero.

Hence, is looks as if the blood relation requirement might be too strong. When reviewer suggested that this problem might be eliminated by saying that the importance of a feature is 0 if there exists a graphical model in which the feature is not in the same connected component as the target (note that this is a graphical model and not a causal model). However, this corollary should be theoretically analyzed.

Some additional comments that emerge in the revised paper:
1.	The axioms do not define a unique solution. Indeed, if a function Imp has all three required properties (axioms) then multiplying it by any positive constant would generate another valid feature importance score. It would be nice to add another requirement that will force a unique solution like Shapley Value or MCI.
2.	The proof in the appendix shows that UMFI has the three required properties only when certain assumptions hold on the distribution. However, in the body of the paper, these limitations are not mentioned.
3.	In line 211 it is stated that the proofs are presented in Section 3, however, the proofs are presented only in the appendix


**Award:**

No

---

### Decision · Program_Chairs · 2022-09-14

Reject